# *Panax Ginseng* alleviates thioacetamide-induced liver injury in ovariectomized rats: Crosstalk between inflammation and oxidative stress

**Rasha E. Mostafa**[1]*, **Nermeen M. Shaffie**[2], **Rasha M. Allam**[1]

**1** Department of Pharmacology, Medical Research and Clinical Studies Institute, National Research Centre, Cairo, Egypt, **2** Department of Pathology, Medical Research and Clinical Studies Institute, National Research Centre, Cairo, Egypt

* re.mostafa@nrc.sci.eg, dr_rosha81@yahoo.com

**Data Availability Statement:** All relevant data are within the manuscript and its Supporting Information files.

## Abstract

Liver diseases impose a substantial health problem. Female hormones play a crucial role in the protection against chronic inflammatory diseases. Fifty female rats were allocated into five groups (n = 10). Group I comprised sham-operated rats. The remaining groups underwent ovariectomy at the beginning of the experiment. Group II served as the ovariectomy-control group. Groups III, IV & V received thioacetamide (TAA; 300 mg/kg; i.p.) to induce liver injury 6 weeks after ovariectomy. Group III served as the TAA-control group. Groups IV & V received *panax ginseng* (100 and 300 mg/kg/day, p.o.) for 6 weeks post TAA administration. All groups were investigated for liver function tests along with total antioxidant capacity (TAC), tumor necrosis factor-α (TNF-α) and advanced glycation end products (AGEs). Histopathological examination of liver tissues was performed followed by immunohistochemical staining for nuclear factor kappa-B (NF-kβ **p65**) and myeloperoxidase (MPO). Ovariectomized-rats showed a non-significant change in the measured parameters while TAA administration resulted in significant liver damage. *Panax ginseng* at both dose levels significantly improved the serum liver function tests and TAC along with decreasing the AGEs and TNF-α. It also restored the histopathological picture of liver tissue and decreased hepatic tissue inflammation via reduction of MPO and NF-kβ **p65** immunoreactivity. The current study is the first to elucidate the effect of *panax ginseng* against TAA-induced liver injury in ovariectomized rats which mimic aged post-menopausal estrogen-deficient females. The study demonstrates the crosstalk between AGEs, NF-kβ and MPO in the modulation of inflammation. *Panax ginseng* possesses antioxidant and anti-inflammatory properties.

## Introduction

Liver diseases impose a substantial global health problem as they are ordinarily associated with significant morbidity and mortality. Hepatocyte death is the major feature of liver disease. Scar

**Funding:** The author(s) received no specific funding for this work.

**Competing interests:** The authors have declared that no competing interests exist.

tissue gradually replaces functional hepatocytes eventually leading to impairment of blood flow and loss of liver functionality [1]. Consequently, inflammatory cytokines and apoptotic markers are significantly initiated [2]. Furthermore, the pro-oxidant and anti-oxidant imbalance leads to lipid peroxidation and massive disturbance of cellular functions. The condition is further complicated by poor regeneration of dying hepatocytes and the building up of collagen fibers within the hepatic tissue [3]. Liver diseases occur due to acute or chronic injuries of the hepatic tissue. Chronic liver disease ultimately results in fibrosis, cirrhosis, and hepatocellular carcinoma. Common causes of liver disease include chronic viral hepatitis, excessive alcohol ingestion, as well as nonalcoholic fatty liver disease [4].

Female hormones play a crucial role in the protection against chronic inflammatory diseases. 17b-estradiol (E2) is the most secreted endogenously synthesized ovarian estrogen [5]. Estrogen is a steroid hormone that controls differentiation and growth in several tissues, including the liver [6]. Reductions in circulating estrogen levels that occur during menopause foster the conditions that increase the damage in the liver tissue. The Beneficial effects of estrogen in liver diseases include enhancement of innate immunity and antioxidant effects along with the promotion of mitochondrial function and DNA synthesis in hepatocytes. Estrogen also causes inhibition of fibrogenesis as well as reduction of the release of oxidative stress parameters and inflammatory cytokines from Kupffer cells [7]. The beneficial activities of estrogen are mainly mediated in the liver via the estrogen receptors ERα. ERα is expressed in about 40% of human hepatocytes as well as mouse hepatocytes. Therefore, post-menopausal females might display more severe liver disease progression [8].

Thioacetamide (TAA) is a potent selective hepatotoxic agent that causes fibrosis and cirrhosis [1, 9]. TAA is used to induce liver injury in experimental animals because it results in excessive accumulation of extracellular matrix proteins as well as induction of apoptosis, inflammation, necrosis, oxidative damage in rat liver. Such events mimic morphological and biochemical alterations observed in human liver injuries [10, 11].

For thousands of years, *Panax ginseng* (Araliaceae), also known as Korean ginseng or Asian ginseng has traditionally been used as a functional food or herbal medicine especially in East Asia [12]. *Panax ginseng* exhibits anti-inflammatory, anti-diabetic, anti-dyslipidemic, anti-tumor and radical-scavenging properties [13]. It also shows protective activities against chronic fatigue together with several cardiovascular, gastrointestinal, and aging-related disorders [14].

The present study aims at inspecting whether *panax ginseng* exerts significant protective effects against thioacetamide-induced liver injury in ovariectomized rats highlighting its potential anti-inflammatory and anti-oxidant properties.

## Materials and methods

### Animals

Adult female Wistar rats weighing 180–200 g were used in the current study. Standard food pellets and tap water were supplied ad libitum. Animals and food pellets were obtained from the animal house colony of the National Research Center (NRC, Egypt). The study was conducted in accordance with the ARRIVE guidelines and following the recommendations of the National Institutes of Health Guide for Care and Use of Laboratory Animals (NIH Publications No. 8023, revised 1978). Moreover, the National Research Centre–Medical Research Ethics Committee (NRC-MREC) for the use of animal subjects approved this study; approval number 1410782021.

## Drugs and chemicals

Thioacetamide (TAA) was purchased from Sigma (St. Louis, MO, USA). TAA was prepared freshly by dissolving in sterile saline and stirred well until all crystals were dissolved. *Panax ginseng* powder root extract 3.5% (Ginseng®, 100 mg soft gelatin capsules; PHARCO pharmaceuticals, Egypt) was used throughout the study. All other chemicals were of the highest analytical grade available.

## Experimental design and treatment protocol

Fifty rats were randomly allocated into five groups, 10 animals each. Group I included sham-operated rats. The remaining 4 groups underwent ovariectomy at the beginning of the experiment. Rats in group II received only saline (2 mL/kg/day, i.p.) for 12 weeks; starting 6 weeks after the ovariectomy and served as ovariectomy-control (OVX-control) group. Groups III, IV & V received TAA (300 mg/once, i.p.) to induce hepatic injury 6 weeks after the ovariectomy. Group III received only saline (2 mL/kg/day, i.p.) for 6 weeks; starting the next day after TAA administration and served as TAA-control group. Groups IV & V received *panax ginseng* (100 and 300 mg/kg/day, p.o.) [15, 16] respectively for 6 weeks; starting the next day after TAA administration. Along the experimental period, animals received dextrose water and ringer lactate solutions (10 mg/kg/day, i.p.) to prevent renal failure, hypoglycemia and electrolyte imbalance [1]. All rats were sacrificed under anesthesia 24h after the last treatment and overnight fasting (**Fig 1**).

## Methods

### Ethics statement

The study was conducted in accordance with the ARRIVE guidelines and following the recommendations of the National Institutes of Health Guide for Care and Use of Laboratory Animals (NIH Publications No. 8023, revised 1978). Moreover, the National Research Centre–Medical Research Ethics Committee (NRC-MREC) for the use of animal subjects approved this study; approval number 1410782021.

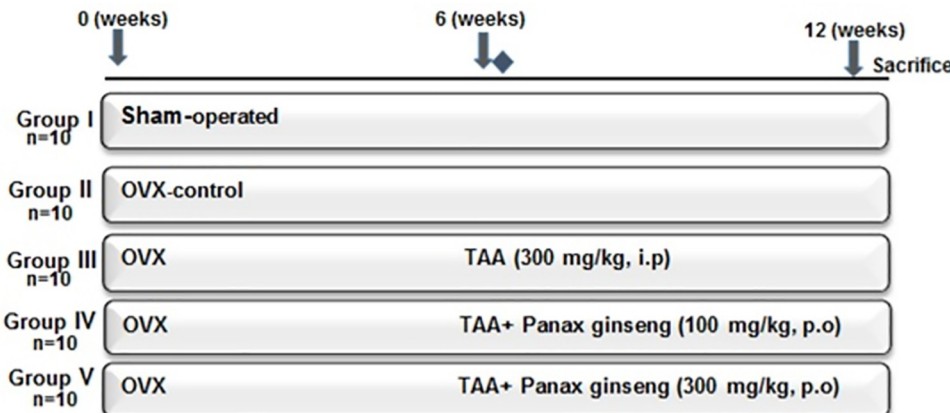

**Fig 1. Schematic diagram of experimental design and treatment protocol.** Rats in sham-operated and OVX-control groups received only saline (2 mL/kg/day, i.p.) for 12 weeks. Groups IV & V received *panax ginseng* for 6 weeks; starting the next day after TAA administration. All animals were sacrificed under anesthesia 24 h after the last drug administration after overnight fasting.

## Ovariectomy

Rats were anesthetized with thiopental sodium (20 mg/kg; ip) and then ovariectomized according to the method described by Turner et al. (2001) [17]. Drug administration was started 6 weeks after ovariectomization and continued for another 6 weeks.

## Biochemical analysis

Twenty-four hours after the last drug dose, rats were anesthetized, and blood samples were withdrawn from the retro-orbital venous plexus. Collected blood samples were allowed to stand for 10 min at room temperature then centrifuged at 4˚C using a cooling centrifuge (Laborezentrifugen, 2k15, Sigma, Germany) at 3000 r.p.m for 10 min and sera were separated for the biochemical assessment. Livers were then harvested from each rat and parts were homogenized (using MPW–120 homogenizer, Med instruments, Poland) to obtain 20% homogenate; the homogenate was centrifuged using a cooling centrifuge (Laborezentrifugen, 2k15, Sigma, Germany) at 3000 r.p.m for 10 min; the supernatant was taken for the tissue bio-chemical analyses. Serum estradiol (E2) was measured using commercially available rat ELISA kits according to the manufacturer's instructions (CUSABIO®, USA, Catalog Number. CSB-E05110r). Serum liver function tests including aspartate aminotransferase (AST), alanine aminotransferase (ALT), alkaline phosphatase (ALP) activities as well as serum levels of total bilirubin & total proteins along with serum and hepatic tissue total antioxidant capacity (TAC) were measured using commercially available kits according to the manufacturer's instructions (Biodiagnostic®, Egypt). Moreover, serum & hepatic tissue concentration of advanced glycation end products (AGEs) and tumor necrosis factor-alpha (TNF-α) were measured using commercially available rat ELISA kits according to the manufacturer's instructions (CUSA-BIO®, USA, Catalog Number. CSB-E09511r) and (Sun-Long Biotech Co. LTD®, China; Catalogs Number. SL0722Ra) respectively.

## Histopathological examination

Other parts of the livers were rinsed with PBS to remove excess blood and then fixed in 10% neutral buffered formalin and embedded in paraffin wax. 4μm thick sections were stained with Hematoxylin and Eosin (H&E) and examined using binocular Olympus CX31 microscope [18].

## Detection of the fibrobasts and collagen fibers

The obtained tissue sections were collected on glass slides, deparaffinized, stained by Masson Trichrom stain and examined through the CX31light electric microscope [18]. The severity of histopathological alterations has been scored by a professional pathologist.

**Immunohistochemical determination of nuclear factor kappa B p65 (NF-kβ p65) and myeloperoxidase (MPO).** Paraffin-embedded liver sections were deparaffinized and hydrated. Immunohistochemistry was performed with Anti-NF-kB p65 antibody (ab16502) and Anti-Myeloperoxidase antibody (ab208670) (Abcam, UK) as primary antibodies. The paraffin sections were heated in a microwave oven (25 min at 720 W) for antigen retrieval and incubated with either anti- NF-kβ p65 or anti-MPO antibodies (1:50 dilution) overnight at 4˚C. After washing with PBS, followed by incubation with biotinylated goat-anti-rabbit-immunoglobulin G secondary antibodies (1:200 dilution; Dako Corp.) and streptavidin/alkaline phosphatase complex (1:200 dilution; Dako) for 30 min at room temperature, the binding sites of antibody were visualized with DAB (Sigma). After washing with PBS, the samples were counterstained with H&E for 2–3 min. and dehydrated by transferring them through

increasing ethanol solutions (30%, 50%, 70%, 80%, 95%, and 100% ethanol). Following dehydration, the slices were soaked twice in xylene at room temperature for 5 min, mounted, examined, and evaluated by a high-power light microscope.Relying on the percentage of positive cells per HPF, the immune reactivity was semi-quantitatively assessed in ten random HPF, according to the method of Hassan et al. (2019), in which; 0 = no staining,1 = positive staining in < 30% of cells per HPF, 2 = positive staining in 30–70% of cells per HPF, or 3 = positive staining in > 70% of cells per HPF [19].

## Statistical analysis

All the values are presented as means ± standard error of the means (SEM). Comparisons between different groups were carried out using one-way analysis of variance (ANOVA) followed by *Tukey's* multiple comparison post hoc test. Results of histopathological and immunohistochemical assessment were analyzed by Kruskal-Wallis non-parametric ANOVA test followed by Mann-Whitney *U* test. The difference was considered significant when *p <0.05*. GraphPad prism® software (version 6 for Windows, San Diego, California, USA) was used to carry out these statistical tests.

## Results

### Effects of *panax ginseng* on serum estradiol (E2) and serum liver function tests in thioacetamide-induced liver injury in ovariectomized rats

Ovariectomization resulted in a significant reduction in serum estradiol (E2) as compared to sham-operated rats (Table 1).

Serum liver function tests were non-significantly elevated in ovariectomized rats when compared to sham-operated rats. Thioacetamide (TAA; 300 mg/kg, i.p.) resulted in

Table 1. Effects of *panax ginseng* on serum estradiol (E2) levels and liver function tests in TAA-induced liver injury in ovariectomized rats.

| Groups \ Parameters | Sham-operated | OVX-control | OVX + TAA-Control (300 mg/kg, i.p.) | OVX+TAA+*panax ginseng* (100mg/kg, p.o.) | OVX+TAA+*Panax ginseng* (300mg/kg, p.o.) |
|---|---|---|---|---|---|
| Estradiol (pg/ml) | 66.60 ± 1.90 | 6.32[a] ± 0.77 | 6.77[a] ± 0.85 | 7.33[a] ± 0.97 | 7.28[a] ± 0.83 |
| AST (IU/L) | 41.60 ± 2.16 | 51.03 ± 2.06 | 159.29[a, b] ± 3.81 | 127.51[a, b,c] ± 4.42 | 102.86[a, b,c] ± 1.84 |
| ALT (IU/L) | 43.87 ± 1.32 | 54.02 ± 1.50 | 166.05[a, b] ± 10.41 | 137.25[a, b,c] ± 2.43 | 111.45[a, b,c] ± 5.75 |
| ALP (IU/L) | 85.73 ± 2.06 | 88.18 ± 3.22 | 139.66[a,b] ± 3.48 | 113.85[a,b,c] ± 4.23 | 98.42[c] ± 2.78 |
| Total Bilirubin (mg/dL) | 0.56 ± 0.02 | 0.62 ± 0.03 | 1.63[a, b] ± 0.04 | 1.32[a, b,c] ± 0.03 | 1.13[a, b,c] ± 0.05 |
| Total protein (g/dL) | 10.33 ± 0.13 | 9.73 ± 0.16 | 5.36[a, b] ± 0.17 | 6.70[a, b,c] ± 0.15 | 7.74[a, b,c] ± 0.17 |

Group I includes sham-operated rats. All other groups underwent ovariectomy at the beginning of the experiment. Rats in the OVX-control group received only saline. The remaining 3 groups received TAA (300 mg/once, i.p.) to induce hepatic injury 6 weeks after the ovariectomy. Groups IV & V received *panax ginseng* (100 and 300 mg/kg/day, p.o.) respectively for 6 weeks; starting the next day after TAA administration. All animals were sacrificed 24 h after the last drug administration.

Data is presented as mean ± S.E. (n = 10). Data was analyzed by one-way ANOVA followed by *Tukey's* post hoc test

[a] significantly different at p<0.05 *vs* sham-operated group

[b] significantly different at p<0.05 *vs* OVX-control group and

[c] significantly different at p<0.05 *vs* OVX + TAA-Control group.

liver injury in ovariectomized rats as evidenced by the significant elevation of serum activities of aspartate transaminase (AST), alanine transaminase (ALT), and alkaline phosphatase (ALP) to 312%, 307% and 158% respectively as compared to the ovariectomy-control group. TAA also resulted in a significant reduction of serum levels of total proteins to 55% as well as significant elevation of serum levels of total bilirubin to 263% as compared to the ovariectomy-control group.

*Panax Ginseng* (100 & 300 mg/kg/day, p.o) significantly decreased the elevated serum AST activity to 80% and 65%, decreased the elevated serum ALT activity to 83% and 67% and decreased the elevated serum ALP activity to 82% and 70%, respectively as compared to the TAA-control group.

*Panax Ginseng* (100 & 300 mg/kg/day, p.o) significantly elevated the reduced serum concentration of total proteins to 125% and 144%. The elevated serum bilirubin was reduced to 81% and 69% as compared to the TAA-control group (Table 1).

## Effects of *panax ginseng* on serum and hepatic tissue Total Antioxidant Capacity (TAC) in thioacetamide-induced liver injury in ovariectomized rats

Serum and hepatic tissue total antioxidant capacity (TAC) were non-significantly decreased in ovariectomized rats as compared to sham-operated rats. Thioacetamide (TAA; 300 mg/kg, i. p.) resulted in a significant reduction in serum and hepatic tissue TAC in ovariectomized rats to 65% and 66% respectively of the ovariectomy-control group.

*Panax Ginseng* (100 mg/kg/day, p.o) significantly elevated the reduced serum and hepatic tissue TAC to 125% and 118% respectively of TAA-control group while *Panax Ginseng* (300 mg/kg) significantly elevated the reduced serum and hepatic tissue TAC to 131% and 127% respectively of TAA-control group (Fig 2).

(a)

(b)

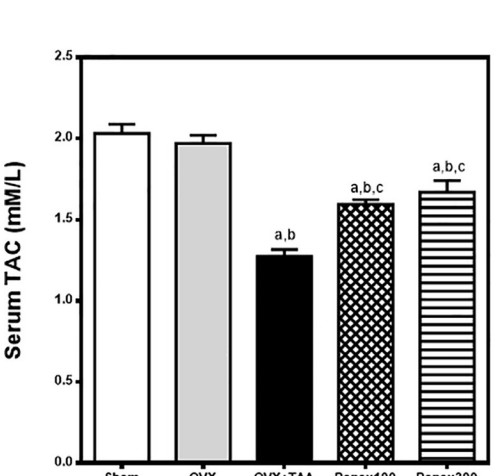
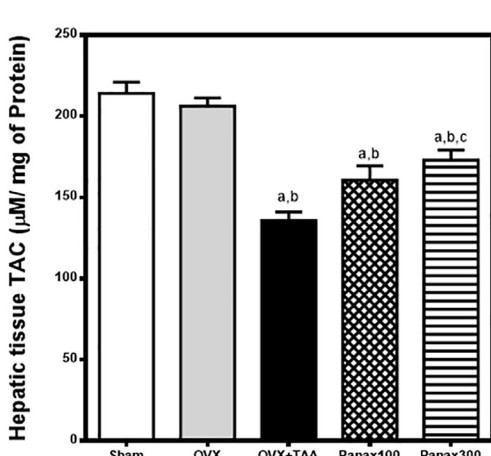

**Fig 2.** Effects of *panax ginseng* on (a) serum & (b) hepatic tissue TAC in TAA-induced liver injury in ovariectomized rats. Group I includes sham-operated rats. All other groups underwent ovariectomy at the beginning of the experiment. Rats in the OVX-control group received only saline. The remaining 3 groups received TAA (300 mg/once, i.p.) to induce hepatic injury 6 weeks after the ovariectomy. Groups IV & V *received panax ginseng* (100 and 300 mg/kg/day, p.o.) respectively for 6 weeks; starting the next day after TAA administration. All animals were sacrificed 24 h after the last drug administration. Data is presented as mean ± S.E. (n = 10). Data was analyzed by one-way ANOVA followed by *Tukey's* post hoc test, [a] significantly different at p<0.05 *vs* sham-operated group, [b] significantly different at p<0.05 *vs* OVX-control group and [c] significantly different at p<0.05 *vs* OVX + TAA-Control group.

### Effects of *panax ginseng* on serum and hepatic tissue advanced glycation end products (AGEs) in thioacetamide-induced liver injury in ovariectomized rats

Serum and hepatic tissue advanced glycation end products (AGEs) were non-significantly elevated in ovariectomized rats as compared to sham-operated rats. Thioacetamide (TAA; 300 mg/kg, i.p.) resulted in a significant elevation in serum and hepatic tissue AGEs in ovariectomized rats to 241% and 274% respectively of the ovariectomy-control group.

*Panax Ginseng* (100 mg/kg) significantly reduced the elevated serum and hepatic tissue AGEs to 84% and 78% respectively of TAA-control group. On the other hand, *Panax Ginseng* (300 mg/kg) significantly reduced the elevated serum and hepatic tissue AGEs to 57% and 67% respectively of TAA-control group (**Fig 3**).

### Effects of *panax ginseng* on serum and hepatic tissue tumor necrosis factor-alpha (TNF-α) in thioacetamide-induced liver injury in ovariectomized rats

Serum and hepatic tissue tumor necrosis factor-alpha (TNF-α) contents were non-significantly elevated in ovariectomized rats as compared to sham-operated rats. Thioacetamide (TAA; 300 mg/kg, i.p.) resulted in a significant elevation in serum and hepatic tissue TNF-α contents in ovariectomized rats to 217% and 175% respectively of the ovariectomy-control group.

*Panax Ginseng* (100 mg/kg) significantly reduced the elevated serum and hepatic tissue TNF-α contents to 80% and 79% respectively of TAA-control group while *Panax Ginseng* (300 mg/kg) significantly reduced the elevated serum and hepatic tissue TNF-α contents to 74% and 70% respectively of TAA-control group (**Fig 4**).

### Histopathological examination of liver tissues

Liver sections from sham-operated rats showed a normal histological structure of the central vein and surrounding hepatocytes. No histopathological alterations could be seen (**Fig 5A**),

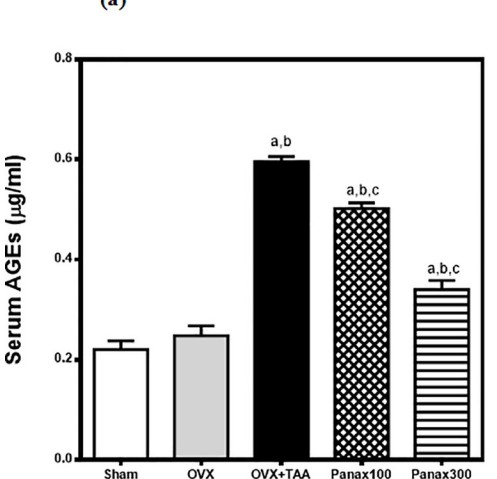
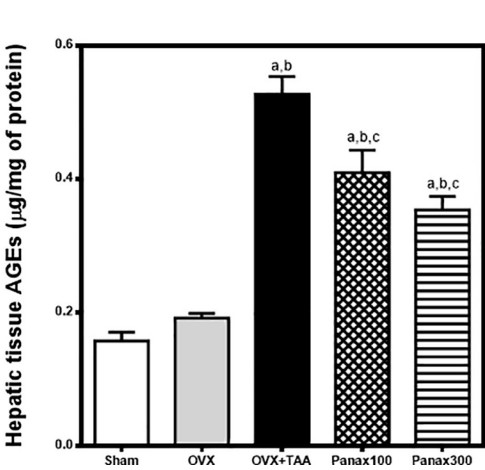

**Fig 3.** Effects of *panax ginseng* on (a) serum & (b) hepatic tissue AGEs in TAA-induced liver injury in ovariectomized rats. Group I includes sham-operated rats. All other groups underwent ovariectomy at the beginning of the experiment. Rats in the OVX-control group received only saline. The remaining 3 groups received TAA (300 mg/once, i.p.) to induce hepatic injury 6 weeks after the ovariectomy. Groups IV & V received *panax ginseng* (100 and 300 mg/kg/day, p.o.) respectively for 6 weeks; starting the next day after TAA administration. All animals were sacrificed 24 h after the last drug administration. Data is presented as mean ± S.E. (n = 10). Data was analyzed by one-way ANOVA followed by *Tukey's* post hoc test, [a] significantly different at p<0.05 *vs* sham-operated group, [b] significantly different at p<0.05 *vs* OVX-control group and [c] significantly different at p<0.05 *vs* OVX + TAA-Control group.

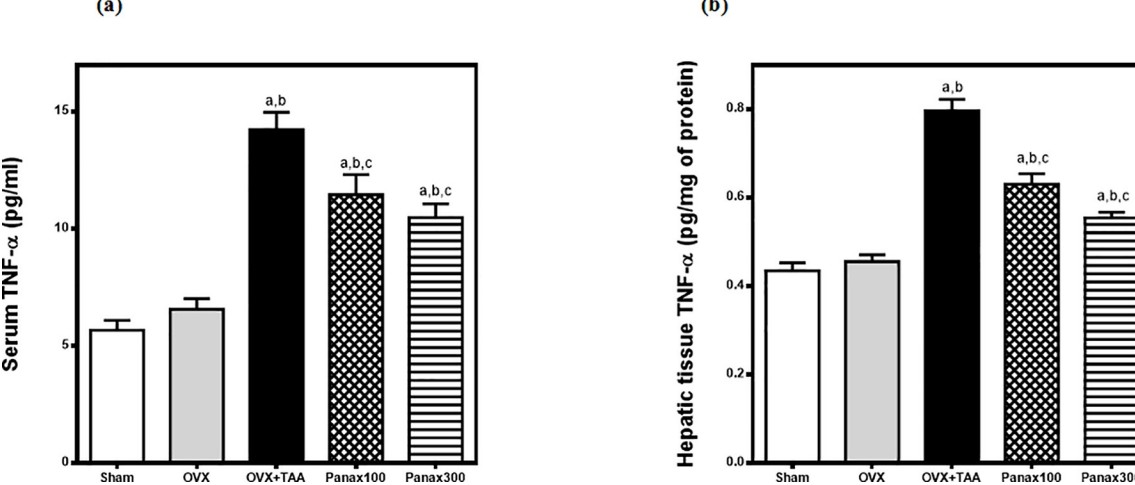

**Fig 4.** Effects of *panax ginseng* on (a) serum & (b) hepatic tissue TNF-α contents in TAA-induced liver injury in ovariectomized rats. Group I includes sham-operated rats. All other groups underwent ovariectomy at the beginning of the experiment. Rats in the OVX-control group received only saline. The remaining 3 groups received TAA (300 mg/once, i.p.) to induce hepatic injury 6 weeks after the ovariectomy. Groups IV & V received *panax ginseng* (100 and 300 mg/kg/day, p.o.) respectively for 6 weeks; starting the next day after TAA administration. All animals were sacrificed 24 h after the last drug administration. Data is presented as mean ± S.E. (n = 10). Data was analyzed by one-way ANOVA followed by *Tukey's* post hoc test, [a] significantly different at $p < 0.05$ *vs* sham-operated group, [b] significantly different at $p < 0.05$ *vs* OVX-control group and [c] significantly different at $p < 0.05$ *vs* OVX + TAA-Control group.

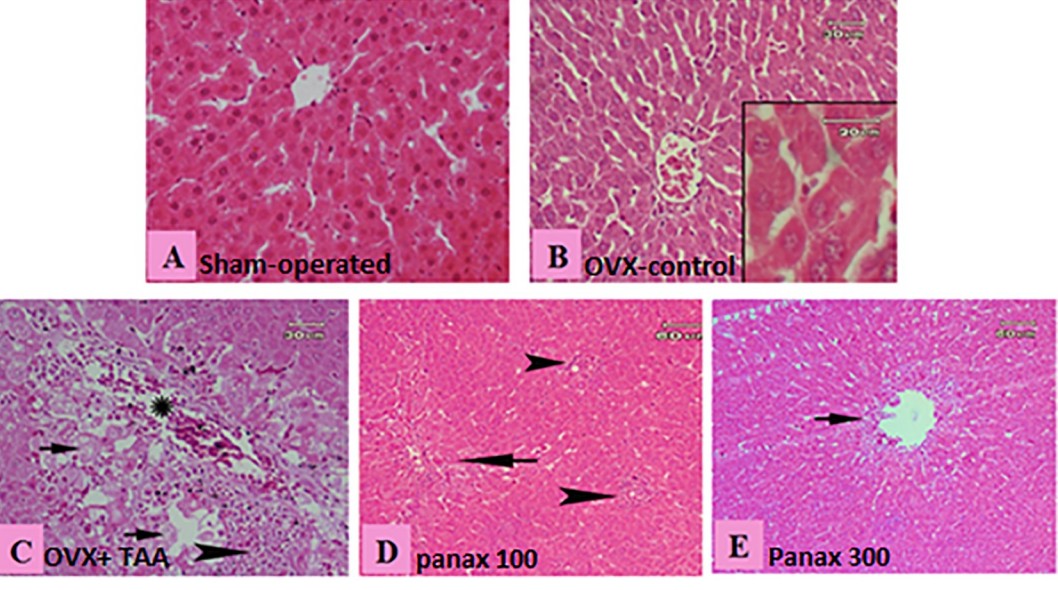

(Stain: H&E, scale bar=100μm).

**Fig 5. Histopathological examination of liver tissue.** Liver section from **(A)** sham-operated rats showing the normal histological structure, **(B)** ovariectomy-control rats showing slight histopathological hepatocyte alterations, **(C)** TAA-control rat showing noticeable fibrosis, vacuolar degeneration and karyolysis in many hepatocytes (arrow) along with aggregations of inflammatory cells (arrowhead) and congestion of main blood vessels (star), **(D)** rats treated with *panax ginseng* (100 mg/kg) showing small necrotic area (arrow) and small focal aggregation of inflammatory cells (arrowhead), and **(E)** rats treated with *panax ginseng* (300 mg/kg) showing healthy hepatocytes and no fibrosis except around the central vein (arrow) **(Stain: H&E).**

while the OVX-control rats showed slight histopathological alterations of the hepatocytes (**Fig 5B**). The TAA-control rats showed noticeable fibrosis in portal area with focal aggregation of inflammatory cells and around the central vein. Vacuolar degeneration and karyolysis could be seen clearly in many hepatocytes along with aggregations of inflammatory cells and congestion of main blood vessels (**Fig 5C**). Rats treated with the low dose of *panax ginseng* (100 mg/kg) showed mild improvement of the histopathological picture as manifested by a small necrotic area and small focal aggregation of inflammatory cells (**Fig 5D**). Moreover, rats treated with the high dose *of panax ginseng* (300 mg/kg) showed a huge improvement of the histopathological picture where no fibrosis could be seen all over the tissue or in between the lobules with healthy hepatocytes except around the central vein (**Fig 5E**).

## Detection of the fibrobasts and collagen fibers using masson trichrome stain

No histopathological alterations were recorded in liver sections from sham-operated rats (**Fig 6A**). OVX-control rats showed slight histopathological alterations of the hepatocytes (**Fig 6B**). The TAA-control rats showed noticeable fibrosis, collagen deposition and centrilobular necrosis distributed in diffuse manner all over the hepatocytes in the parenchyma and in portal area (**Fig 6C**). Rats treated with the low dose of *panax ginseng* (100 mg/kg) showed moderate Centrilobular necrosis of the hepatic parenchyma. Fibroblasts could be seen in a diffuse manner associated with congestion in the portal vein along with infiltration of inflammatory cells in the portal area (**Fig 6D**). Moreover, rats treated with the high dose of *panax ginseng* (300 mg/

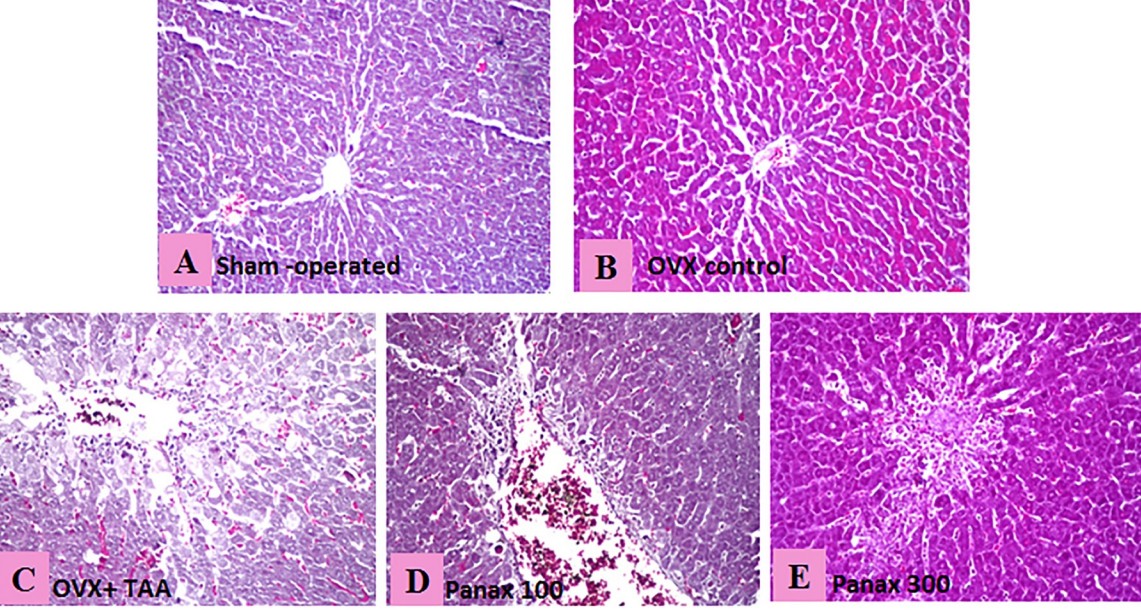

**Stain: Masson Trichrome**

**Fig 6. Histopathological detection of the fibrobasts and collagen fibers.** Liver section from (**A**) sham-operated rats showing the normal histological structure, (**B**) ovariectomy-control rats showing slight histopathological hepatocyte alterations, (**C**) TAA-control rat showing noticeable fibrosis, collagen deposition and centrilobular necrosis in diffuse manner all over the hepatic parenchyma, (**D**) rats treated with *panax ginseng* (100 mg/kg) showing moderate fibrosis of the hepatic parenchyma. Fibroblasts could be seen in a diffuse manner associated with a congestion in the portal vein along with infiltration of inflammatory cells in the portal area, and (**E**) rats treated with *panax ginseng* (300 mg/kg) showing a significant improvement of the histopathological picture where fibroblasts and collagen fibers could scarcely be noticed all over the tissue. (**Stain: Masson Trichrome**).

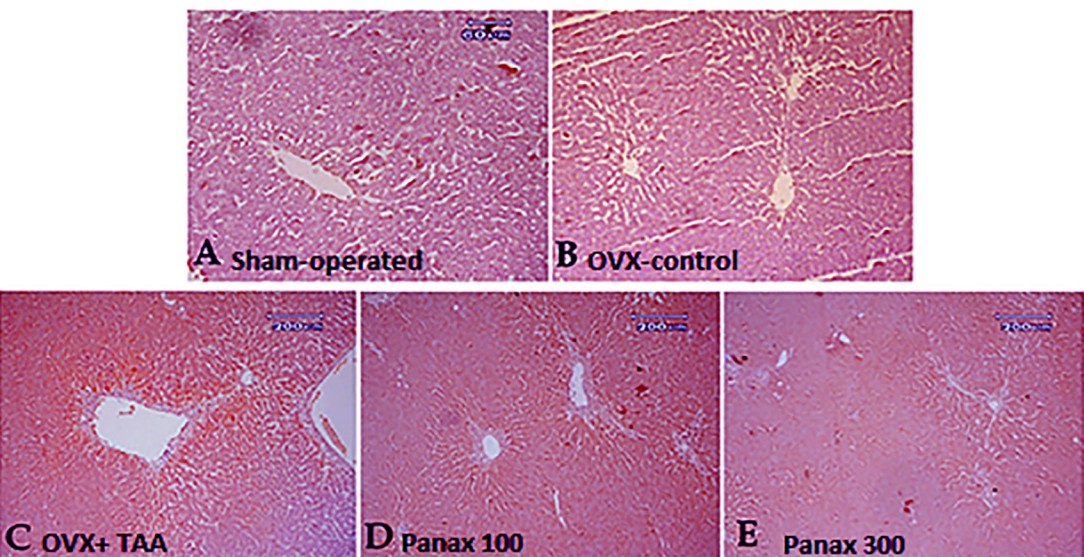

(NF-kβ immunohistochemical staining, scale bar=100µm)

**Fig 7. Immunohistochemical examination of NF-kβ p65 in liver tissue.** Liver tissues, immunohistochemically stained with anti-NF-kβ **p65** antibodies, from **(A)** sham-operated rat, **(B)** ovariectomy-control rat showing no expression of NF-kβ in the liver tissue, **(C)** TAA-control rat showing the strong positive result of the stain especially around the central veins, **(D)** rat treated with *panax ginseng* (100 mg/kg) showing a mild decrease in positively stained cells, and **(E)** rat treated with *panax ginseng* (300 mg/kg) showing a remarkable decrease in the positive result **(NF-kβ p65 immunohistochemical staining)**.

kg) showed a significant improvement of the histopathological picture where fibroblasts and collagen fibers could scarcely be noticed all over the tissue. Centrilobular necrosis was not detected in the hepatic lobules all over the parenchyma (**Fig 6E**).

## Immunohistochemical determination of NF-kβ p65

Liver sections from sham-operated rats and OVX-control rats stained immunohistochemically with anti-NF-kβ **p65** antibodies showed negative results for the stain.

The TAA-control rats showed a strong positive result of the stain especially around the central veins. Rats treated with *panax ginseng* (100 mg/kg) showed a mild improvement as manifested by a decrease in positively stained cells. Furthermore, rats treated with *panax ginseng* (300 mg/kg) showed a massive improvement a remarkable decrease in the positive result (**Fig 7**). The semi-quantitative scoring of the immune reactivity of NF-kβ p65 in liver tissues of normal and treated rats is presented in Table 2.

## Immunohistochemical determination of MPO

Liver sections from sham-operated rats and OVX-control rats stained immunohistochemically with anti-MPO antibodies showed negative results for the stain.

The TAA-control rats showed a robust positive result of the stain all over the lobule. The treatment with 100 mg/kg of *panax ginseng* resulted in mild improvement as demonstrated by a decrease in positively stained cells. Additionally, the high dose of *panax ginseng* (300 mg/kg) displayed a vast amelioration with a substantial decrease in the positive result (**Fig 8**). The semi-quantitative scoring of the immune reactivity of MPO in liver tissues of normal and treated rats is presented in Table 3.

**Table 2. The scoring of histopathological detection of the fibrobasts and collagen fibers recorded in the liver tissue of normal and treated rats.**

| Groups | Fibrous Connective tissue poliferation | Fibroblasts and collagen fibers | Congestion in portal vein | Inflammatory cells infiltration | Centriloblar necrosis |
|---|---|---|---|---|---|
| Sham-operated | - | - | - | - | - |
| OVX-control | + | + | - | - | - |
| OVX + TAA-Control (300 mg/kg, i.p.) | +++ | +++ | +++ | +++ | +++ |
| OVX+TAA+panax ginseng (100mg/kg, p.o.) | ++ | ++ | ++ | + | + |
| OVX+TAA+panax ginseng (300mg/kg, p.o.) | + | + | - | - | - |

+++ denotes Severe alteration, ++ denotes Moderate alteration, + denotes Mild alteration &–denotes Nil.

## Discussion

Hormonal involvement has been overly linked to liver diseases in postmenopausal women. The multiple and diverse physiological as well as biochemical alterations that occur during menopause greatly affect liver functionality and may promote liver disease [7].

In the current work, ovariectomization of rats significantly decreased serum estradiol (E2) levels in comparison to the sham-operated group, signifying reproductive senescence of animals in response to ovariectomization. Similar to our work, Sunar *et al.* (2009) reported a significant reduction in serum estrogen levels in the ovariectomized groups as compared to the control rats [20].

Moreover, ovariectomization of rats resulted in non-statistically significant changes in liver functions accompanied by a decrement in serum and hepatic tissue TAC, an elevation of serum and hepatic tissue AGEs & TNF-α along with histopathological abnormalities in comparison to the sham-operated group. These abnormalities are mainly attributed to the lack of estrogen's hepato-protection. Estrogen has been proven to be very crucial to hepatocyte functionality as it promotes mitochondrial function, cellular immunity, and antioxidant capacity. Estrogen also inhibits fibrogenesis together with cellular senescence [21]. Menopause is characterized by gradual reproductive aging accompanied by a deficiency of circulating estrogen. Menopausal women show high susceptibility to liver pathology especially nonalcoholic fatty liver diseases, fibrosis, cirrhosis, hepatocellular cancers and ultimately liver failure [22]. Menopausal changes and estrogen deficiency result in mitochondrial dysfunction. Numerous studies reported that estrogen receptors are found in the hepatic mitochondria. Mitochondrial dysfunction leads to disrupted cell membrane permeability, cellular aging with the profound cessation of cellular growth and eventually cellular death [23]. Morphological and histological abnormalities are also apparent in livers of postmenopausal versus premenopausal women. Such abnormalities include a gradual and persistent decline in liver volume, blood flow and regeneration capacity [24]. Moreover, an abrupt decline in liver antioxidant capacity occurs in menopause. Deficiency in antioxidant enzymes in livers of postmenopausal women leads to decreased elimination of reactive oxygen species, reduced protection against oxidative insults and subsequently elevation in lipid peroxidation [25]. Interestingly, estrogen deficiency has also been linked to the potentiation of inflammatory processes in different bodily organs [26].

It is noteworthy that ovariectomization showed no effect on TAA- induced acute liver failure in rats. Koblihova *et al.* (2020) reported that the degree of liver injury and the survival rate were comparable in ovariectomized rats administered with TAA versus intact non-

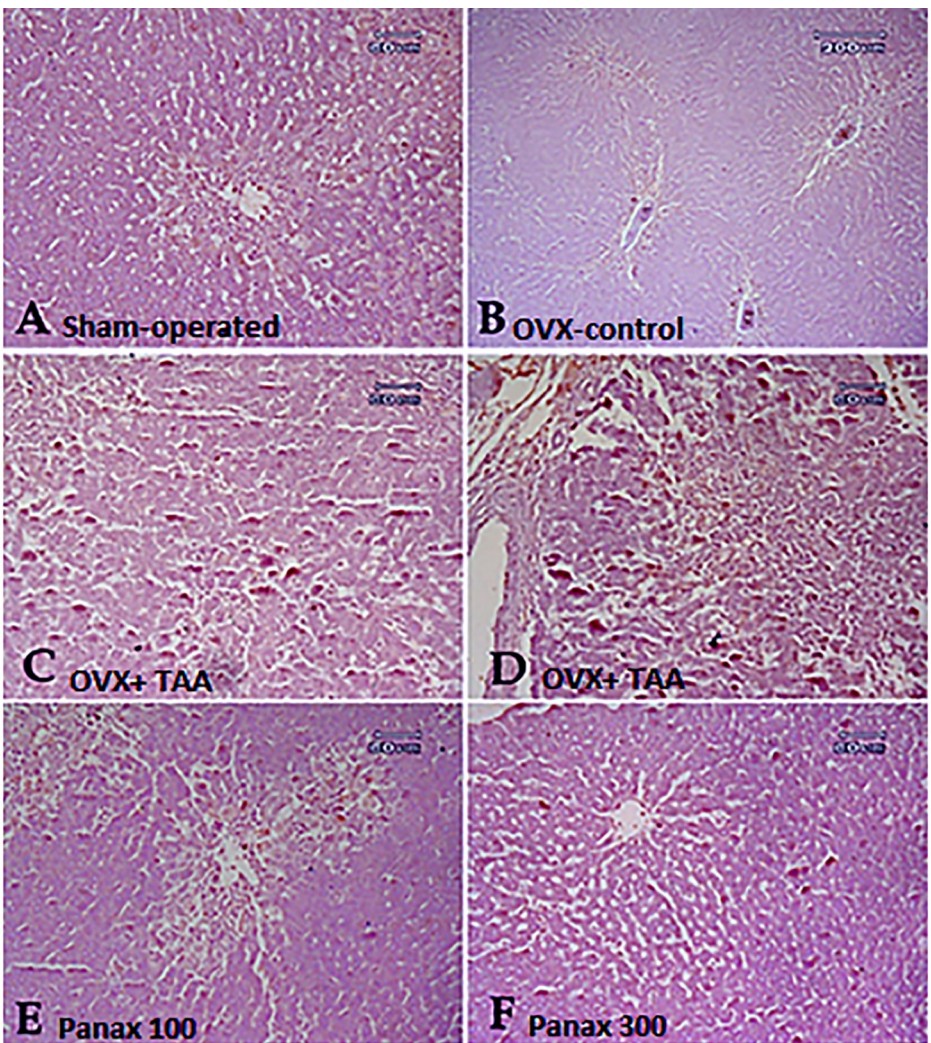

## (MPO immunohistochemical staining, scale bar=100μm)

**Fig 8. Immunohistochemical examination of MPO in liver tissue.** Liver tissues, immunohistochemically stained with anti-MPO antibodies, from **(A)** sham-operated rats, **(B)** ovariectomy-control rats didn't show notable expression of NF-KB in the liver tissue, **(C, D)** TAA-control rats showing a robust positive result of the stain all over the lobule, **(E)** rats treated with *panax ginseng* (100 mg/kg) showing a mild decrease in the positively stained cells, and **(F)** rats treated with *panax ginseng* (300 mg/kg) showing a remarkable decrease in the positive result (**MPO immunohistochemical staining**).

ovariectomized female rats [27]. In the present study, ovariectomized rats treated with TAA showed a significant elevation of AST, ALT, ALP, total bilirubin, AGEs and TNF-α accompanied by a significant decrement in total protein and TAC in comparison to ovariectomized control rats. Moreover, OVX+ TAA rats showed abundant histopathological changes and increased immunohistochemical staining of inflammatory markers; NF-kβ **p65** and MPO. Of note, TAA administration caused a significant fibrosis in hepatic tissue, accompanied by fibrous connective tissue proliferation and deposition of fibroblasts, collagen fibers as well as inflammatory cells all over the hepatic parenchyma. TAA is reported to cause significant injury in hepatic tissue thus affecting the liver's functions. TAA also causes hepatic oxidative stress, inflammation and eventually fibrosis [28]. Lee *et al.* (2019) reported that estrogen deficiency

**Table 3. The results of NF-kβ p65 and MPO expression recorded in the liver tissue of normal and treated rats.**

| Groups | NF-kβ p65 | MPO |
|---|---|---|
| | (% of positive cells/HPF) | (% of positive cells/HPF) |
| Sham-operated | 0.00[d] ± 0.00 | 0.00[d] ± 0.00 |
| OVX-control | 0.00[d] ± 0.00 | 0.00[d] ± 0.00 |
| OVX + TAA-Control (300 mg/kg, i.p.) | 2.70[a] ± 0.20 | 2.80[a] ± 0.17 |
| OVX+TAA+*panax ginseng* (100mg/kg, p.o.) | 1.70[b] ± 0.25 | 1.30[b] ± 0.29 |
| OVX+TAA+*panax ginseng* (300mg/kg, p.o.) | 0.90[c] ± 0.30 | 0.60[c] ± 0.21 |

Immunohistochemical staining assessment was analyzed by performing Kruskal-Wallis nonparametric ANOVA test followed by Mann-Whitney U test. Different lowercase letters are significantly different (p< 0.05).

potentiated the severity of TAA-induced hepatic fibrosis in rats due to increased oxidative stress and inflammation [8].

We investigated the anti-inflammatory and antioxidant effects of *panax ginseng* against TAA-induced liver injury in ovariectomized rats. *Panax ginseng* contains many active constituents like polysaccharides, fatty acids and ginsenosides. *Panax ginseng* is now widely available in the market in many pharmaceutical dosage forms. *Panax ginseng* possesses diverse biological actions including anti-diabetic, anti-hyperlipidemic, neuro-protective, antineoplastic, immune-modulating as well as hepatoprotective effects [4]. The beneficial effects of *panax ginseng* on liver functionality have been proven against various liver diseases. Kim *et al.* (1997) stated that *panax ginseng* protects against carbon tetrachloride-induced liver injury [29]. Wu *et al.* (2001) reported that *panax ginseng* protects against diethylnitrosamine-induced hepatocellular carcinoma [30], and Abdel-Wahhab *et al.* (2011) reported that it has beneficial actions against viral hepatitis [31].

Moreover, Lee *et al.* (2015) and Seo *et al.* (2013) reported that *panax ginseng* exhibits beneficial actions against alcohol-induced hepatotoxicity [32, 33]. Hong *et al.* (2013) demonstrated hepatoprotective actions of *panax ginseng* against non-alcoholic fatty liver disease [34]. Lee and Oh (2015) showed that *panax ginseng* improves aging-related learning & memory deficiencies [35]. However, to the author's knowledge, this is the first study to elucidate the effect of *panax ginseng* against TAA-induced liver injury in ovariectomized rats which mimic aged post-menopausal estrogen-deficient females. Most of *panax ginseng*'s effects are mainly due to its anti-oxidant and anti-inflammatory properties. In the current work, the use of *panax ginseng* elevated the concentration of TAC and decreased the concentration of AGEs and TNF-α in serum and hepatic tissues. Oxidative stress leads to tissue damage and subsequently initiates inflammatory reactions. Kim *et al.* (2017) reported that *panax ginseng* (100 mg/kg) restored antioxidant components in liver cells, including TAC [13]. Oxidative stress fosters the formation of AGEs. AGEs are heterogeneous moieties, formed *in vivo* via oxidative as well as non-oxidative interactions between sugars and their protein and lipid adducts. The aging process, diabetes and malignancy accelerate the formation & accumulation of AGEs and activate receptors for AGEs (RAGE) [36]. Increased circulating levels of AGEs and activation of RAGE induce inflammatory responses in different cells including hepatocytes. Oxidative stress also initiates cellular inflammatory reactions [37]. It is noteworthy that AGEs are mainly metabolized and cleared via the liver. Therefore, increasing AGEs concentration creates a positive feedback loop where AGEs harm the liver function. Thus, AGEs metabolism and clearance

will be decreased leading to their accumulation. This further hurts liver tissues; eventually leading to boosting of circulating AGEs [38]. Few studies reported the beneficial effects of *panax ginseng* on the amelioration of AGEs in different tissues. Tan *et al.* (2015) demonstrated that *panax ginseng* enhances cognitive function in rats via reduction of AGEs [39]. Quan *et al.* (2013) reported that *panax ginseng* relieves AGEs-mediated kidney injury in rats [40]. However, this is the first study to pinpoint *panax ginseng*'s ameliorative effects on AGEs after induction of liver injury with TAA especially in the case of ovariectomization and estrogen deficiency.

In the current work, *panax ginseng* also alleviated the histopathological changes in liver tissues and decreased the immunohistochemical staining of the inflammatory markers, NF-kβ and MPO. This emphasizes *panax ginseng*'s anti-inflammatory potential. It is reported that the binding of AGEs to RAGE triggers oxidative stress as well as inflammatory signaling pathways via activation of NF-kβ and its downstream pathway via elevation of serum & hepatic tissue TNF-α concentrations. NF-kβ is a transcription factor that initiates and modulates inflammatory and immune responses via the release of numerous cytokines and chemokines [41]. TNF-α is a pleiotropic cytokine initiating various inflammatory actions via NF-kβ signaling pathways, especially in the hepatic tissue [42]. There is a significant crosstalk between NF-kβ signaling and MPO. MPO plays a crucial role in linking inflammation and oxidative stress [43]. Pan *et al.* (2018) reported the release of MPO from macrophages when triggered by NF-kβ [44]. Ethridge *et al.* (2004) demonstrated an inhibition in MPO activity as a result of reduction in NF-kβ concentration in pancreatic tissues [45]. In-line with our data, Bae *et al.* (2014) reported that *panax ginseng* decreased gastric inflammation via reduction of MPO levels in Mongolian gerbils [46]. Park *et al.* (2009) showed that *panax ginseng*'s anti-inflammatory properties might be attributed to the reduction in NF-kβ concentrations [47]. Tan *et al.* (2015) demonstrated that *panax ginseng* treatment caused resulted in a reduction of percentage of NF-kβ immunohistochemical staining in cortical and hippocampal tissues [39]. Hou *et al.* (2014) reported that *panax ginseng* treatment significantly reduced the elevated TNF-α in hepatic tissue post carbon tetrachloride administration in rats [48].

The study suggests adding *panax ginseng* to the treatment protocol for liver diseases in post-menopausal women. Thus, we recommend more in-depth molecular studies on *panax ginseng*'s anti-inflammatory effects based on the present findings. Also we recommend assessment of the potential anti-fibrotic properties of *Panax ginseng* in future research.

## Conclusion

Taken together, the current study is the first to elucidate the effect of *panax ginseng* against TAA-induced liver injury in ovariectomized rats which mimic aged post-menopausal estrogen-deficient females. *Panax ginseng* possesses anti-oxidant as well as anti-inflammatory properties. *Panax ginseng* reduces circulating AGEs and thus spares liver tissues and improves liver functionality. It also improves the liver's histopathological pictures and reduces hepatic tissue inflammation via reduction of MPO and NF-kβ **p65** immunohistochemical staining. TNF-α; the downstream factor for NF-kβ signaling pathway; has also been reduced in response to *panax ginseng* treatment. The study demonstrates the crosstalk between AGEs, NF-kβ, TNF-α and MPO in modulation of inflammation.

## Supporting information

**S1 Data.**
(XLSX)

**S1 File.**
(DOC)

**S1 Graphical abstract.**
(TIF)

## Author Contributions

**Conceptualization:** Rasha E. Mostafa, Rasha M. Allam.

**Data curation:** Rasha E. Mostafa, Nermeen M. Shaffie.

**Investigation:** Rasha E. Mostafa, Nermeen M. Shaffie.

**Methodology:** Nermeen M. Shaffie, Rasha M. Allam.

**Writing – original draft:** Rasha E. Mostafa.

**Writing – review & editing:** Rasha E. Mostafa, Rasha M. Allam.

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
