## [Decision Letter · Decision Letter 0]

1 Oct 2021

PONE-D-21-25160Panax Ginseng Alleviates Thioacetamide-Induced Liver Injury in Ovariectomized Rats Via Modulation of Advanced Glycation End Products: Crosstalk between Inflammation and Oxidative StressPLOS ONE

Dear Dr. Mostafa,

Apologies for the delay in the revision of your manuscript, due to the difficulties in finding reviewers in summer time.Thank you for submitting your manuscript to PLOS ONE. After careful consideration, we feel that it has merit but does not fully meet PLOS ONE’s publication criteria as it currently stands. Therefore, we invite you to submit a revised version of the manuscript that fully addresses the points raised by the reviewer.

We look forward to receiving your revised manuscript.

Kind regards,

Matias A Avila, Ph.D.

Academic Editor

PLOS ONE

Journal Requirements:

Reviewers' comments:

Reviewer's Responses to Questions

**Comments to the Author**

1. Is the manuscript technically sound, and do the data support the conclusions?

Reviewer #1: Yes

2. Has the statistical analysis been performed appropriately and rigorously? 

Reviewer #1: Yes

3. Have the authors made all data underlying the findings in their manuscript fully available?

Reviewer #1: Yes

4. Is the manuscript presented in an intelligible fashion and written in standard English?

Reviewer #1: Yes

5. Review Comments to the Author

Reviewer #1: In this work Mostafa et al evaluate the protective effects of Panax ginseng extract on thioacetamide (TAA) induced liver injury in ovariectomized rats, which mimic post-menopausal estrogen-deficient females. The authors find significant hepatoprotection at different levels, including antioxidant liver capacity, generation of AGEs, inflammation and injury. This study confirms the hepatoprotective and beneficial effects of Panax ginseng observed in previous works, including ovariectomized rats. There are some issues that need the authors attention.

1. It appears that the authors administered a single TAA injection, and evaluated liver injury-related parameters six weeks after this single dose. It is hard to believe that six weeks after that single TAA injection there are still signs of liver injury detectable. Upon acute liver injury rodents trigger a potent regenerative response. Other works using similar single doses of TAA report a rapid recovery of serum liver enzymes and bilirrubin, which almost normalize after one week (see for instance: PMID: 32901492). This aspect is critical and should be clarified.

2. A recent study (Kobilhova E, et al. Physiol. Res. 2020: PMID: 32901492) demonstrated the lack of effect of ovariectomy on TAA-induced acute liver injury in rats. This previous publication is directly relevant to the present report and should be cited and discussed.

3. In Fig. 5 the authors discuss about the degree of liver fibrosis found in tissue sections from the different groups of rats. Fibrosis needs to be directly assessed, at least by Sirius Red staining, and quantitated.

4. In Fig. 6, immunohistochemistry for Nf-KB should also be quantitated. The Nf-KB subunit examined and the reference of the commercial antibody used should be provided.

5. In Fig. 7, immunohistochemistry for MPO should also be quantitated. The reference of the commercial antibody used should be provided.

6. The title should be modified, no direct proof that Panax ginseng alleviates TAA-induced liver injury via AGEs modulation is provided. The authors only show a correlation betweed this treatment and a reduction in AGEs levels, as they also find a correlation with increased hepatic antioxidant capacity.

Minor.

“Panax ginseng” should be always written in italics.

6. PLOS authors have the option to publish the peer review history of their article (what does this mean?). If published, this will include your full peer review and any attached files.

Reviewer #1: No

---

## [Author Response · Author response to Decision Letter 0]

5 Nov 2021

Response to reviewers’ comments

Rebuttal letter

Manuscript number: PONE-D-21-25160.

Title: Panax Ginseng Alleviates Thioacetamide-Induced Liver Injury in Ovariectomized Rats Via Modulation of Advanced Glycation End Products: Crosstalk between Inflammation and Oxidative Stress.

We sincerely thank the editor and the respectable reviewers for their precious time and constructive criticisms. Their valuable comments were of great help in revising the manuscript and they definitely improve it. 

Accordingly, the revised manuscript has been systematically enhanced with new information and 

additional interpretations. Following are the detailed responses to the journal requirements and 

reviewers comments:

Please find below a point to point response to the reviewers’ comments:

All changes are marked in the main manuscript file via the track changes mode.

Comments to the authors

Journal requitements have been strictly followed

All data will be uploaded as one single EXCEL file into the online system

Data Availability statement has been changed in the cover letter.

 Reviewers' comments:

Reviewer's Responses to Questions

Comments to the Author

1. Is the manuscript technically sound, and do the data support the conclusions?

Reviewer #1: Yes

2. Has the statistical analysis been performed appropriately and rigorously?

Reviewer #1: Yes

3. Have the authors made all data underlying the findings in their manuscript fully available?

Reviewer #1: Yes

4. Is the manuscript presented in an intelligible fashion and written in standard English?

Reviewer #1: Yes

5. Review Comments to the Author

Reviewer #1:

In this work Mostafa et al evaluate the protective effects of Panax ginseng extract on thioacetamide (TAA) induced liver injury in ovariectomized rats, which mimic post-menopausal estrogen-deficient females. The authors find significant hepatoprotection at different levels, including antioxidant liver capacity, generation of AGEs, inflammation and injury. This study confirms the hepatoprotective and beneficial effects of Panax ginseng observed in previous works, including ovariectomized rats. There are some issues that need the authors attention.

1. It appears that the authors administered a single TAA injection, and evaluated liver injury-related parameters six weeks after this single dose. It is hard to believe that six weeks after that single TAA injection there are still signs of liver injury detectable. Upon acute liver injury rodents trigger a potent regenerative response. Other works using similar single doses of TAA report a rapid recovery of serum liver enzymes and bilirrubin, which almost normalize after one week (see for instance: PMID: 32901492). This aspect is critical and should be clarified.

We would like to thank the reviewer for this question

Many research papers demonstrated that the use of single oral dose of TAA resulted in significant liver injury that last for more than 4 weeks post TAA adminstration.

Mostafa et al. (2017) demonstrated that a single intraperitoneal dose of TAA (300 mg/kg) resulted in significant hepatotoxic oxidative damage in rats after 30 days post TAA injection. The Histopathological examination of liver tissues showed significant hepatotoxicity as manifested by severe congestion of blood vessels, activated Kupfer cells, and enlarged hepatocytes with disappearance of sinusoids (Mostafa et al., 2016).

Waters et al. (2005) reported that TAA causes centrilobular liver necrosis following single dose administration, while repeated exposure results in bile duct proliferation along with hepatic cirrhosis (Waters et al., 2005).

Strnad et al. (2008) reported that TAA caused more severe liver injury and fibrosis in mice than CCL-4 (Strnad et al., 2008).

Of note, Wallace et al. (2015) reported that TAA is commonly used in the dose of 100-200 mg/kg, i.p. to induce long-lasting reproducible liver fibrosis that resembles alcoholic fibrogenesis with low mortality in experimental animals (Wallace et al., 2015).

The administration of TAA via the oral or IP route is a safe and effective technique for the study of fibrosis in rodents. There is a wide variation in the route, dose, timing and animal used (Wallace et al., 2015).

Mostafa, R.E., Salama, A.A., Abdel-Rahman, R.F., Ogaly, H.A., 2016. Hepato-and neuro-protective influences of biopropolis on thioacetamide-induced acute hepatic encephalopathy in rats. Canadian Journal of Physiology and Pharmacology 95, 539-547.

Strnad, P., Tao, G.Z., Zhou, Q., Harada, M., Toivola, D.M., Brunt, E.M., Omary, M.B., 2008. Keratin Mutation Predisposes to Mouse Liver Fibrosis and Unmasks Differential Effects of the Carbon Tetrachloride and Thioacetamide Models. Gastroenterology 134, 1169-1179.

Wallace, M., Hamesch, K., Lunova, M., Kim, Y., Weiskirchen, R., Strnad, P., Friedman, S., 2015. Standard operating procedures in experimental liver research: thioacetamide model in mice and rats. Laboratory animals 49, 21-29.

Waters, N.J., Waterfield, C.J., Farrant, R.D., Holmes, E., Nicholson, J.K., 2005. Metabonomic deconvolution of embedded toxicity: application to thioacetamide hepato-and nephrotoxicity. Chemical research in toxicology 18, 639-654.

Like wise in the current work, all measured parameters showed significant deterioration 6 weeks post TAA dose. The Histopathological examination of liver tissues showed noticeable fibrosis in portal area with focal aggregation of inflammatory cells and around the central vein. Vacuolar degeneration and karyolysis could be seen clearly in many hepatocytes along with aggregations of inflammatory cells and congestion of main blood vessels. 

2. A recent study (Kobilhova E, et al Res. 2020: PMID: . Physiol. 32901492) demonstrated the lack of effect of ovariectomy on TAA-induced acute liver injury in rats. This previous publication is directly relevant to the present report and should be cited and discussed.

We sincerely thank the reviewer for pointing to this study.

Actually our data is in-line with this work since ovariectomized rats showed a non-significant change in the measured parameters

The aforementioned study has been cited and discussed in the revised manuscript.

3. In Fig. 5 the authors discuss about the degree of liver fibrosis found in tissue sections from the different groups of rats. Fibrosis needs to be directly assessed, at least by Sirius Red staining, and quantitated.

Thank you for this valuable comment.

Liver fibrosis has been assessed and quantified as per the reviewer comment.

Masson trichrome staining has been used for this assessment since Sirius Red staining is not available in our lab.

Masson trichrome staining is frequently and accurately used for assessment of fibrosis.

This data is added in figure 6 and the quantified results have been added in table 2. 

4. In Fig. 6, immunohistochemistry for Nf-KB should also be quantitated. The Nf-KB subunit examined and the reference of the commercial antibody used should be provided.

We sincerely thank the reviewer for this valuable remark

The Nf-KB subunit examined (Anti-NF-kB p65) and the reference of the commercial antibody (ab16502, Abcam, UK) have been added to materials and methods section in page 8.

Moreover, the immunohistochemistry for Nf-KB p65 was quantitated and added in results section in table 3

5. In Fig. 7, immunohistochemistry for MPO should also be quantitated. The reference of the commercial antibody used should be provided.

Like wise, the reference of the commercial antibody for MPO (ab208670, Abcam, UK) have been added to materials and methods section in page 8.

Moreover, the immunohistochemistry for MPO was quantitated and added in results section in table 2

6. The title should be modified, no direct proof that Panax ginseng alleviates TAA-induced liver injury via AGEs modulation is provided. The authors only show a correlation betweed this treatment and a reduction in AGEs levels, as they also find a correlation with increased hepatic antioxidant capacity.

We appreciate this comment so the title has been modified as suggested by the reviewer to be “Panax Ginseng Alleviates Thioacetamide-Induced Liver Injury in Ovariectomized Rats: Crosstalk between Inflammation and Oxidative Stress.”

7. Minor.

“Panax ginseng” should be always written in italics.

Thank you for this comment.

Panax ginseng has been changed into italics allover the manuscript as per the reviewer’s comment.

Thank you

---

## [Editor Report · Decision Letter 1]

11 Nov 2021

Panax Ginseng Alleviates Thioacetamide-Induced Liver Injury in Ovariectomized Rats : Crosstalk between Inflammation and Oxidative Stress.

PONE-D-21-25160R1

Dear Dr. Mostafa,

We’re pleased to inform you that your manuscript has been judged scientifically suitable for publication and will be formally accepted for publication once it meets all outstanding technical requirements.

Kind regards,

Matias A Avila, Ph.D.

Academic Editor

PLOS ONE
---

## [Editor Report · Acceptance letter]

15 Nov 2021

PONE-D-21-25160R1 

*Panax Ginseng* Alleviates Thioacetamide-Induced Liver Injury in Ovariectomized Rats: Crosstalk between Inflammation and Oxidative Stress. 

Dear Dr. Mostafa:

I'm pleased to inform you that your manuscript has been deemed suitable for publication in PLOS ONE. Congratulations! Your manuscript is now with our production department. 

Kind regards, 

on behalf of

Dr Matias A Avila 

Academic Editor

PLOS ONE